# Quantum sensing of microRNAs with nitrogen-vacancy centers in diamond
Justas Zalieckas [1] ✉, Martin M. Greve[1], Luca Bellucci[2,3], Giuseppe Sacco[4], Verner Håkonsen [5], Valentina Tozzini [2,3,6] & Riccardo Nifosì [2,3] ✉

Label-free detection of nucleic acids such as microRNAs holds great potential for early diagnostics of various types of cancers. Measuring intrinsic biomolecular charge using methods based on field effect has been a promising way to accomplish label-free detection. However, the charges of biomolecules are screened by counter ions in solutions over a short distance (Debye length), thereby limiting the sensitivity of these methods. Here, we measure the intrinsic magnetic noise of paramagnetic counter ions, such as $Mn^{2+}$, interacting with microRNAs using nitrogen-vacancy (NV) centers in diamond. All-atom molecular dynamics simulations show that microRNA interacts with the diamond surface resulting in excess accumulation of Mn ions and stronger magnetic noise. We confirm this prediction by observing an increase in spin relaxation contrast of the NV centers, indicating higher $Mn^{2+}$ local concentration. This opens new possibilities for next-generation quantum sensing of charged biomolecules, overcoming limitations due to the Debye screening.

Detection of biomolecules for diagnostics purposes is typically accomplished using fluorescence labels and labor-intensive amplification methods. Achieving label-free detection with high sensitivity would be of great importance for accessible and early diagnosis of diseases such as cancer. A common approach to the problem would be to measure intrinsic charges of biomolecules with field effect transistor biosensors. However, the major drawback of this method is screening of biomolecular charges by counter ions in solutions, referred to as Debye screening[1]. The effect of the Debye screening is described by the corresponding length (or volume) reflecting $1/e$ decrease of electric potential of the biomolecular charges. Under physiological conditions, the Debye length is less than 1 nm, and it has generally been assumed that electronic detection beyond this distance is virtually impossible[2], which, as a result, affects the detection limit of this type of sensors. To overcome this challenge an alternative approach is needed.

Quantum sensing based on atomic-size and negatively charged nitrogen-vacancy (NV) centers in diamond has attracted a vast interest in the scientific community. In addition to their common application for sensing electric[3,4] and magnetic fields[5,6], NV centers were also used for studying proteins[7] and detection of nuclear magnetic resonance[8] and electron spin resonance (ESR)[9]. Furthermore, NVs can provide the sensitivity needed to detect ESR spectra of DNA duplexes labeled with a nitroxide spin label[10], sense SARS-CoV-2[11] and enable ultrasensitive nanodiamond-based detection of HIV virus[12]. Such versatility and superior sensitivity make NVs a promising quantum diagnostics platform, potentially applicable for an early detection of numerous diseases, as well as for a broad spectrum of biomedical applications[13]. A promising biomarker for diagnostics of various types of cancer[14-16], neurogenerative[17] and autoimmune[18] diseases are short RNA molecules referred to as microRNAs. MicroRNAs are small non-coding RNAs, consisting of approximately 20-22 nucleotides, that play an important role in regulation of post-transcriptional gene expression[19] and were shown to have clinical application in the detection of cancer[20], and monitoring of tumorigenesis and metastasis[21]. The conventional strategies for detection and quantification of microRNAs include Northern blotting[22], quantitative reverse transcription polymerase chain reaction[23], and oligonucleotide microarrays[24] with such limitations as poor sensitivity, labor-intensive steps, and requirement of high-precision thermal cycling equipment[25]. We propose that an alternative approach for detecting and analyzing microRNAs could be exploitation of NV centers.

The NV center consists of a substitutional nitrogen and an adjacent vacancy pair, where the symmetry axis can be oriented along four possible {111} crystalline directions. The triplet ground state of the NV center exhibits a zero-field splitting between the $m_s = 0$ state and the (degenerate) $m_s = \pm1$ states. The spin state of the NV center can be manipulated using green laser light (532 nm), efficiently polarizing it into the $m_s = 0$ state, referred to as the bright state, with a polarization efficiency exceeding 90%[26].

[1]Department of Physics and Technology, University of Bergen, Bergen, Norway. [2]Istituto Nanoscienze - CNR, Pisa, Italy. [3]Lab NEST Scuola Normale Superiore, Pisa, Italy. [4]Scuola Internazionale Superiore di Studi Avanzati (SISSA), Trieste, Italy. [5]NTNU NanoLab, Norwegian University of Science and Technology, Trondheim, Norway. [6]Istituto Nazionale di Fisica Nucleare (INFN), sezione Pisa, Pisa, Italy. ✉e-mail: justas.zalieckas@uib.no; riccardo.nifosi@nano.cnr.it

This polarized state, left to evolve, relaxes to thermal equilibrium (mix of $m_s = 0$ and $m_s = \pm 1$ states) with time constant $T_1$.

The longitudinal spin-lattice relaxation time $T_1$ is prone to a high frequency (~GHz) magnetic noise (i.e., a randomly fluctuating magnetic field with zero average) originating close to the NV center inside the diamond due to bulk impurities[27], at the interface due to surface impurities and defects[28,29] or outside the diamond due to paramagnetic $Mn^{2+}$ [30,31], $Gd^{3+}$ [32] or $Fe^{3+}$ [33] ions present on or close to the surface. Positively charged ions such as $Mn^{2+}$ are electrostatically attracted by negatively charged DNA/RNA backbone phosphate groups and are adsorbed by negatively charged diamond surface[30]. Moreover, the concentration of divalent cations near DNA is strongly enhanced compared to monovalent cations (e.g., $Na^+$, $K^+$)[34]. Therefore, Mn ions could mediate the interaction between oligonucleotides and negatively charged diamond surface in a similar way as other divalent ions, such as $Mg^{2+}$, $Ni^{2+}$, $Co^{2+}$ or $Zn^{2+}$ mediate DNA adsorption to mica surface[35,36], with no significant effect on the structure and the integrity of the oligonucleotides[37]. Due to the preferential divalent cations binding compared to monovalent cations, it is expected that $Mn^{2+}$ interaction with oligonucleotides would yield an excess concentration of Mn ions in the "ionic atmosphere" around nucleic acid, thus increasing magnetic noise and reducing NVs relaxation time $T_1$. Such quantum sensing modality can be used to detect the presence of nucleotides close to the diamond surface and probe their interactions with surface functional groups. Therefore, sensing magnetic noise, instead of partially screened charges, would overcome limitations due to the Debye screening.

In this work we use $T_1$ relaxometry to sense microRNAs with NVs in diamond by measuring magnetic noise of ions counteracting microRNAs intrinsic charges. More specifically, we study single stranded microRNA-21 (miR-21), which has been reported to be upregulated in many cancer types such as gliomas[38], breast cancer[15], and colorectal cancer[39]. We observe an increase in spin relaxation contrast of the NV centers due to a higher magnetic noise induced by excess of Mn cations in the presence of miR-21. To better understand miR-21 interactions with counter ions and diamond surface we perform all-atom molecular dynamics simulations confirming experimental observations that presence of miR-21 close to the diamond surface results in excess accumulation of Mn ions. These results open new possibilities for next-generation quantum biosensors based on NV centers in diamond.

## Results

### Sensing of miR-21 with NV centers

The principle of the proposed label-free detection method and the experimental setup are schematically shown in Fig. 1a. We consider an oxygenated diamond surface in contact with a solution containing miR-21 and $Mn^{2+}$ with electronic spin $S = 5/2$. In such a system, Mn ions counteract the negative charge of miR-21 phosphate groups as well as the negative charge arising either from transfer of electrons to the diamond surface or ionization of residual oxygen-containing functional groups[40]. The Mn ions can mediate the adsorption of miR-21 to the diamond surface, leading to accumulation of additional paramagnetic counter ions and stronger external magnetic noise close to NVs. The relation linking this magnetic noise to the longitudinal relaxation rate $\Gamma_1 = 1/T_1$ of the NVs has been derived in previous works[41] and is

$$\Gamma_1 = \frac{1}{T_1} = \frac{1}{T_1^{int}} + 3\gamma_e^2 \langle B_\perp^2 \rangle \frac{\tau_c}{1 + \omega_0^2 \tau_c^2}. \tag{1}$$

Here, $T_1^{int}$ denotes the intrinsic relaxation time due to phonon-induced transitions, spin-spin interactions, and interactions of NVs with the spin bath in the bulk or at the surface[27–29], when paramagnetic ions are absent. $\gamma_e$ is the electron gyromagnetic ratio and $\langle B_\perp^2 \rangle$ is the variance of the transverse magnetic field with respect to the NV center orientation (the longitudinal component does not induce NV spin transitions). The fraction $\tau_c/(1 + \omega_0^2 \tau_c^2)$ comes from considering the spectral component at $\omega_0 = 2\pi D$

($D = 2.87$ GHz is the energy gap between $m_s = 0$ and $m_s = \pm 1$ states of the NV) of a magnetic field described by $\langle B(t)B(t + \tau) \rangle = \langle B^2 \rangle e^{-|\tau|/\tau_c}$, where $\tau_c$, the autocorrelation decay time, is determined by the paramagnetic spin dynamics[30]. The Mn ions in solution, adsorbed to the surface, or interacting with the nucleic acid, all have different spin dynamics, and contribute with different $\tau_c$ to the overall NV relaxation rate $\Gamma_1$. An accurate evaluation of this parameter in all such cases would require a sophisticated approach[42] beyond the scope of this work. However, unless $\tau_c$ differs by several orders of magnitude from $1/\omega_0 = 55$ ps, the contribution in Eq. (1) will be non-vanishing (see Supplementary Fig. S1). The variance $\langle B_\perp^2 \rangle$ can be calculated from the molecular dynamics simulations (see Methods section) and decays as the 6th power of the distance between each Mn ion and the NV center. Clearly, at low $Mn^{2+}$ bulk concentration, accumulation of $Mn^{2+}$ close to the surface will sensibly increase the magnetic noise experienced by the NVs, thereby affecting their longitudinal relaxation rate.

We use electronic grade single crystal diamond sample with NV centers embedded ~7 nm below the {100} surface to sense miR-21. The negatively charged state of the diamond surface is achieved by treating it in a strong oxidizing agent (Piranha solution). The longitudinal relaxation of NVs is probed using pulse sequence shown in the inset of Fig. 1b. Readout and initialization of spins is achieved with the same green laser light (532 nm) pulse[43]. To improve sensitivity, instead of measuring $T_1$ curve, we only measure spin relaxation at time $\tau_1 = 10\ \mu s$ and $\tau_2 = 400\ \mu s$, with both times indicated by the dashed vertical lines in Fig. 1b. The $\tau_1$ is set to 10 μs instead of 0 μs to avoid potential overlap in the pulse sequence due to finite response times of electronic components and acousto-optic modulator. The spin contrast is then estimated from a ratio of two single point measurements at $\tau_1$ and $\tau_2$[44]. All measurements are performed with the diamond sample placed inside the microfluidic device and with no external magnetic field applied, addressing all four crystallographic NV orientations for maximal sensitivity.

Prior to miR-21 sensing measurements, 1 mM ethylenediaminetetraacetic acid (EDTA) solution (pH 2.0) is injected into the microfluidic channel at a flow rate of 150 μL min⁻¹. This serves two purposes: (i) to neutralize the negatively charged diamond surface (see discussion below) and (ii) to chelate paramagnetic ions (note that chelation efficiency drops with solution pH). Figure 1c shows the spin relaxation contrast measurement as a function of time. The red curve in Fig. 1c represents the change in spin relaxation contrast for sequential injection of the following four solutions: (I) 1 mM EDTA, (II) 5 mM $MnCl_2$ and 10 mM NaCl, (III) 1 μM miR-21 in 5 mM $MnCl_2$ and 10 mM NaCl, (IV) 1 mM EDTA. The concentration of $Mn^{2+}$ in stock solutions is chosen such that contribution to $\langle B_\perp^2 \rangle$ from freely diffusing ions is supressed and is mainly dominated by the adsorbates[30]. When solution II is injected into the microfluidic channel at a flow rate of 150 μL min⁻¹ the spin contrast increases by 1.3% (to ~1.04) due to adsorbed Mn ions on the diamond surface. Solution III (pH 5.2) is prepared from solution II by adding 1 μM of miR-21 and is injected into the microfluidic device at a flow rate of 50 μL min⁻¹. We observe a 1.0% (to ~1.05) increase in spin contrast with respect to solution II, demonstrating quantum sensing of miR-21 with NV centers. The increase in the spin contrast is due to a ~1.5 kHz enhancement of the relaxation rate $\Gamma_1$, as estimated from the fluorescence decays reported in Supplementary Information Fig. S2. The higher relaxation is attributed to the interaction of miR-21 with the diamond surface mediated by Mn counter ions. We explain this observation as follows. The negative charge of $-21e$ for each interacting miR-21 is counteracted by adsorbates and paramagnetic ions in the diffuse layer. Therefore, the presence of miR-21 close to the surface results in higher $Mn^{2+}$ concentration and increased $\langle B_\perp^2 \rangle$ relative to solution II. Next, solution IV is injected (same as solution I) at a flow rate of 150 μL min⁻¹. The EDTA chelates Mn ions and removes them from the microfluidic chamber, effectively restoring the initial state of diamond surface[30] and thus reducing the spin contrast to the similar level as for the solution I. The difference in the spin contrast measured for solution IV compared to solution I is attributed to temperature fluctuations (within ~1.5 °C) in the room during the measurements (see Supplementary Fig. S3).

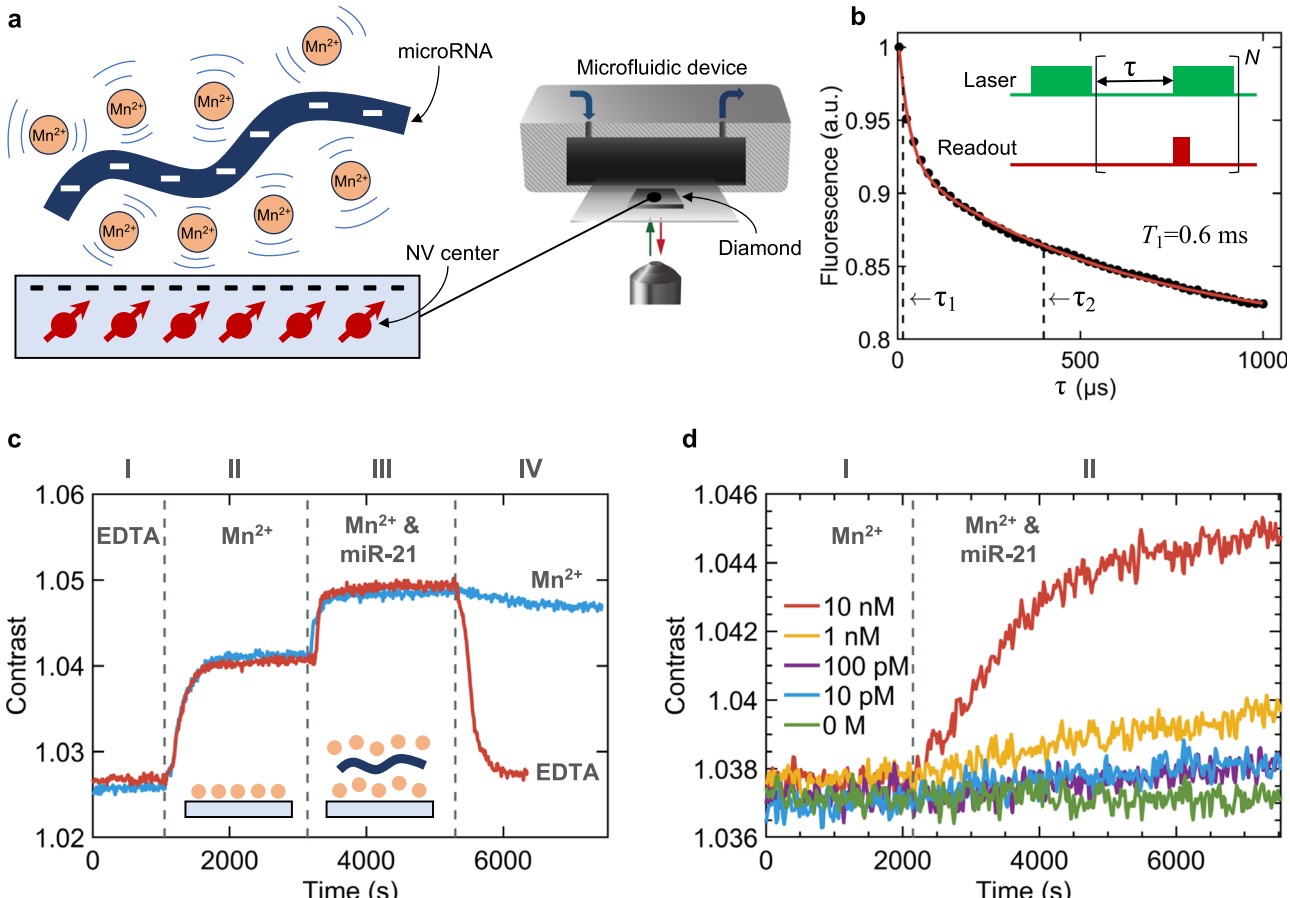

**Fig. 1 | Quantum sensing of microRNAs using NV centers. a** Schematic representation of the experimental setup and the principle of the proposed label-free detection method. Manganese cations ($Mn^{2+}$) counteract intrinsic charge of microRNA-21 (miR-21) and mediate interaction with the negatively charged diamond surface. The NV centers, shown as red circles, are distributed 7±3 nm below the surface. The diamond sample is attached to a coverslip placed inside the microfluidic device, which is used for sequential liquid injection. **b** Measurement of the NVs longitudinal spin-lattice relaxation time ($T_1$) in water. The fluorescence from NVs is measured $N = 20,000$ times for each interrogation time τ using the pulse sequence shown in the inset. The spin contrast is estimated from a ratio of two single point measurements at $\tau_1 = 10$ μs and $\tau_2 = 400$ μs, with both times indicated by the dashed vertical lines. **c** The change in spin relaxation contrast for sequential injection of: (I) 1 mM EDTA, (II) 5 mM $MnCl_2$ and 10 mM NaCl, (III) 1 μM miR-21 in 5 mM $MnCl_2$ and 10 mM NaCl, and (IV) 1 mM EDTA (red line) or 5 mM $MnCl_2$ and 10 mM NaCl (blue line). Vertical dashed lines represent the time of the next solution injection. **d** The change in spin relaxation contrast for sequential injection of: (I) 5 mM $MnCl_2$ and 10 mM NaCl, and (II) miR-21 concentration ranging from 10 pM to 10 nM prepared in the stock solution I. The vertical dashed line represents the time of injection of the solution II. Each data-point represents an average of $N = 40,000$ measurements. All data-points are temperature-corrected to account for temperature fluctuations in the room during the measurements.

The blue line in Fig. 1c shows the change in spin relaxation contrast for sequential injection of the same solutions as described above, except the solution IV was changed to 5 mM $MnCl_2$ and 10 mM NaCl (same as solution II). Removing weakly interacting and freely diffusing miR-21 by introducing solution IV at flow rate of 150 μL min$^{-1}$ only slightly decreases the spin contrast. Since NVs are sensitive to paramagnetic ions 20–30 nm away from diamond surface[30,31] and given ~1.2 nm cross-sectional size of single stranded microRNA we conclude that most of miR-21s are adsorbed to the surface forming a uniform layer. The slight decrease in the spin contrast can be explained by removal of weakly interacting miR-21 and the corresponding Mn counter ions.

To determine the sensitivity of the presented method, we measure the response of the experimental setup to the injection of miR-21 concentrations ranging from 10 pM to 10 nM. Figure 1d shows the change in spin relaxation contrast for sequential injection of: (I) 5 mM $MnCl_2$ and 10 mM NaCl, and (II) miR-21 in 5 mM $MnCl_2$ and 10 mM NaCl. The solution I is injected at a flow rate of 100 μL min$^{-1}$ while the flow rate of the solution II is kept at 150 μL min$^{-1}$. Prior to each measurement the diamond surface was flushed for 1 h with Tris-EDTA buffer solution (pH 8.0) to remove $Mn^{2+}$ and miR-21 adsorbates. To increase the sensitivity and reduce noise, each data point is measured for $N = 40,000$ times for each interrogation time τ.

Since $T_1$ is susceptible to temperature[27], all data-points in Fig. 1d are temperature-corrected (see Methods) to account for temperature fluctuations (within ~1.5 °C) during the measurements. We observe that, upon injection of 10 nM miR-21, the spin contrast rapidly increases with time (red curve in Fig. 1d). After ~4200 s the increase slows down and the contrast approaches the saturation value observed for 1 μM miR-21. This might be attributed to two possible effects: (i) the changes of the local environment such as pH and charge state of the surface due to accumulation of miR-21 adsorbates and higher local $Mn^{2+}$ density or (ii) aggregation of oligonucleotides due to the high ratio of Mn ions to phosphate groups in miR-21[37]. Lowering the concentration of injected miR-21 down to 1 nM and lower yields linear response of the spin contrast. Control measurement with no miR-21 (green curve in Fig. 1d) yields no observable increase of the spin contrast. We observe increase of the spin contrast exceeding 3 standard deviations with respect to the control measurement for 10 pM miR-21 concentration and no observable response of the spin contrast for lower concentrations. Therefore, the limit of detection (LOD) of the presented method is 10 pM for given experimental conditions (Supplementary Fig. S4a shows the calibration curve). Given the microfluidic channel volume of ~12 mm$^3$, the LOD of 10 pM translates to 120 attomoles.

It has been shown that spin relaxation measurements, apart from temperature, can also be susceptible to NV centers charge conversion (induced by laser excitation)[45], diamagnetic electrolyte solutions[46] and electric noise[29,47]. Since the excitation power intensity (9 kW cm$^{-2}$) used is below saturation (~100 kW cm$^{-2}$)[48] and surface oxygenation stabilizes shallow negatively charged NVs[49], effects attributed to NVs charge conversion are neglected in this work. Moreover, concentration of 10 mM of diamagnetic NaCl used in the presented measurements has been shown not to affect $T_1$[46]. Furthermore, we test if charged miR-21 adsorbates induce observable electric noise by measuring the change in spin relaxation contrast with no Mn ions present (see Supplementary Fig. S4b). We observe no measurable change in the spin contrast between 10 mM NaCl solution and 1 µM miR-21 in 10 mM NaCl. This demonstrates that electric noise has negligible effect on the presented measurements.

X-ray photoelectron spectroscopy (XPS) analysis confirms that after Piranha treatment, the diamond surface is functionalized with oxygen groups (see Supplementary Fig. S5a). The observed Si 2 s and Si 2p peaks in the survey spectrum indicate sample contamination with silicon either in the bulk or on the surface. We stress that Si contamination in commercial diamond, which presumably stems from the diamond synthesis, has been reported in several studies in the literature[50,51]. Efforts to improve cleaning and handling of the diamond sample did not suppress the observed signal, suggesting that silicon contamination is distributed in the bulk. The binding energy of the Si 2p peak is also consistent with that of silicon carbide (100–101 eV)[52]. Given the weak signal of these peaks, we expect that silicon contamination does not significantly influence the interaction of Mn ions and miR-21 with the diamond surface. The diamond surface can be negatively charged either due to deprotonated oxygen groups or due to the transfer of electrons to the surface[40]. Figure 2a shows the high-resolution C 1 s XPS spectrum with fitted peaks corresponding to different bonding configuration near the surface. Assuming a complete coverage of oxygenated species (see Supplementary Fig. S5b), the results translate to surface coverage of 64.5% hydroxyl/epoxy (C-OH/C-O-C), 20.2% carbonyl (C=O), 13.7% ether (O-C-O) and 1.6% carboxyl (COOH) functional groups. Only the carboxyl group is deprotonated at pH 5.2 owning to its low pK$_a$ value of ~4.0 (COOH → COO$^-$ + H$^+$)[40]. Since error on the estimation of COOH coverage can be high, we probe the charge state of the diamond surface by measuring the change in spin contrast for solution with pH < 5.2. Figure 2b shows the change in spin contrast for 5 mM MnCl$_2$ solution injected at a flow rate of 150 µL min$^{-1}$ with pH lowered stepwise from 5.3 to 2.0 using hydro-chloric acid. Lowering solution pH decreases the spin contrast, reaching similar value as for the water for pH 2.0. This indicates that Mn ions are desorbed, and the surface charge state is neutral (carboxyl groups are fully protonated) in the solution with pH 2.0. This observation confirms the presence of COOH groups on the surface and verifies XPS results. We note that $T_1$ for nanodiamond modified by carboxyl groups on the particle

surface was found to depend on ambient pH between pH 3 and pH 7[53]. According to that study, $T_1$ shortening at higher pH is due to the electric noise induced by deprotonation of COOH groups. Since in the present work only 1.6% of the diamond surface is covered with COOH groups, NVs relaxation is expected to be dominated by the magnetic noise induced by Mn ion adsorbates.

## Atomic force microscopy and XPS analysis of miR-21 adsorbates

We verify adsorption of miR-21 to the diamond surface using atomic force microscopy (AFM). For this purpose, we compare surface topography of the diamond before and after the adsorption of miR-21. First, we image the surface of the diamond after the Piranha treatment (see Fig. 3a). The image reveals the polishing stripes, clearly seen diagonally across the sample. Next, the sample was covered with miR-21 by treating it in a Solution III (see Methods) and re-imaged (see Fig. 3b) using the same tip and imaging conditions as for the untreated surface. Since the surface roughness is of the same order as the cross-sectional size of miR-21 (~1 nm) and the tip radius is ~5 nm, it is difficult to resolve single miR-21. However, when carefully comparing Fig. 3a, b, it is possible to see that the still present polishing marks appear less pronounced/sharp, and the overall surface is more granular/speckled. To better understand the observed topographical differences, we look in the frequency domain and perform a 2D Fourier transform (2DFFT) of each of the two images (see Fig. 3c, d). We point at four noteworthy features in the 2DFFT images: (1) the bright central spot, corresponding to low frequency, (2) the bright diagonal stripe from lower left to upper right corner emerging due to the polishing stripes on the diamond surface, 3) a set of weaker vertical and horizontal bands passing through the center of the image due to scanning line artefacts, and (4) the diffuse halo, surrounding the bright central spot. The diffuse halo is of particular interest with respect to the surface granularity. First, the finer the granular structure is the larger the diameter of the halo will become, corresponding to the higher frequencies. Second, the intensity of the halo at a given point reflects the occurrence of that signal. Therefore, increase in the diameter of the halo corresponds to occurrence of finer topographical features on the surface. After exposing diamond surface to miR-21 we observe a halo with the larger diameter (Fig. 3d) compared to the one for the untreated surface (Fig. 3c) indicating the presence of miR-21 adsorbates.

To further verify the presence of miR-21 adsorbates we scrape the surface with the AFM tip in a contact mode using a force of ~230 nN. Figure 3e shows the AFM image, acquired in a tapping mode, of the diamond surface topography after scraping the adsorbed miR-21. The average thickness of the removed layer is found to be ~0.5 nm, by measuring the height profile (white line in Fig. 3e) and taking the difference between the average sample heights on either side (blue dashed lines in Fig. 3e). The hight profile is measured parallel to the polishing stripes, away from the aggregated material at the edges. The estimated average thickness is smaller than

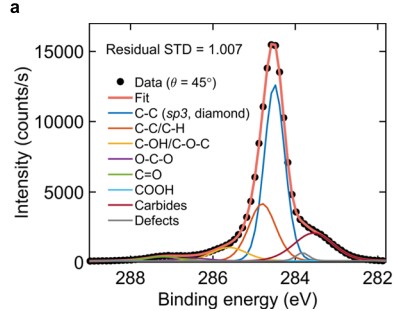
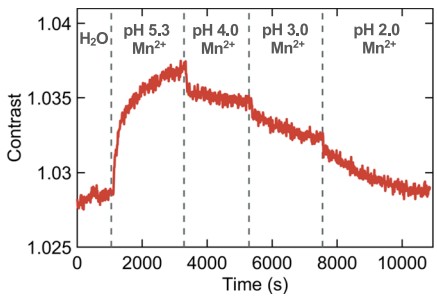
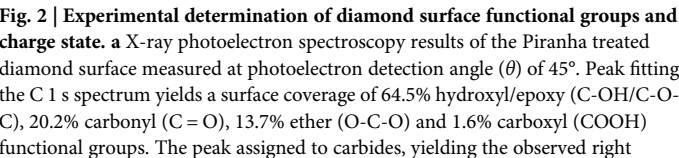

**Fig. 2 | Experimental determination of diamond surface functional groups and charge state. a** X-ray photoelectron spectroscopy results of the Piranha treated diamond surface measured at photoelectron detection angle ($\theta$) of 45°. Peak fitting the C 1 s spectrum yields a surface coverage of 64.5% hydroxyl/epoxy (C-OH/C-O-C), 20.2% carbonyl (C = O), 13.7% ether (O-C-O) and 1.6% carboxyl (COOH) functional groups. The peak assigned to carbides, yielding the observed right

shoulder in the spectrum, comes mainly from Si (C-Si) contamination in the bulk of diamond. The goodness-of-fit is denoted by the residual standard deviation (STD), in which a value of unity corresponds to a perfect fit. **b** The change in spin contrast for 5 mM MnCl$_2$ solution injected at a flow rate of 150 µL min$^{-1}$ with pH lowered stepwise from 5.3 to 2.0 using hydro-chloric acid. Vertical dashed lines represent the time of the next solution injection.

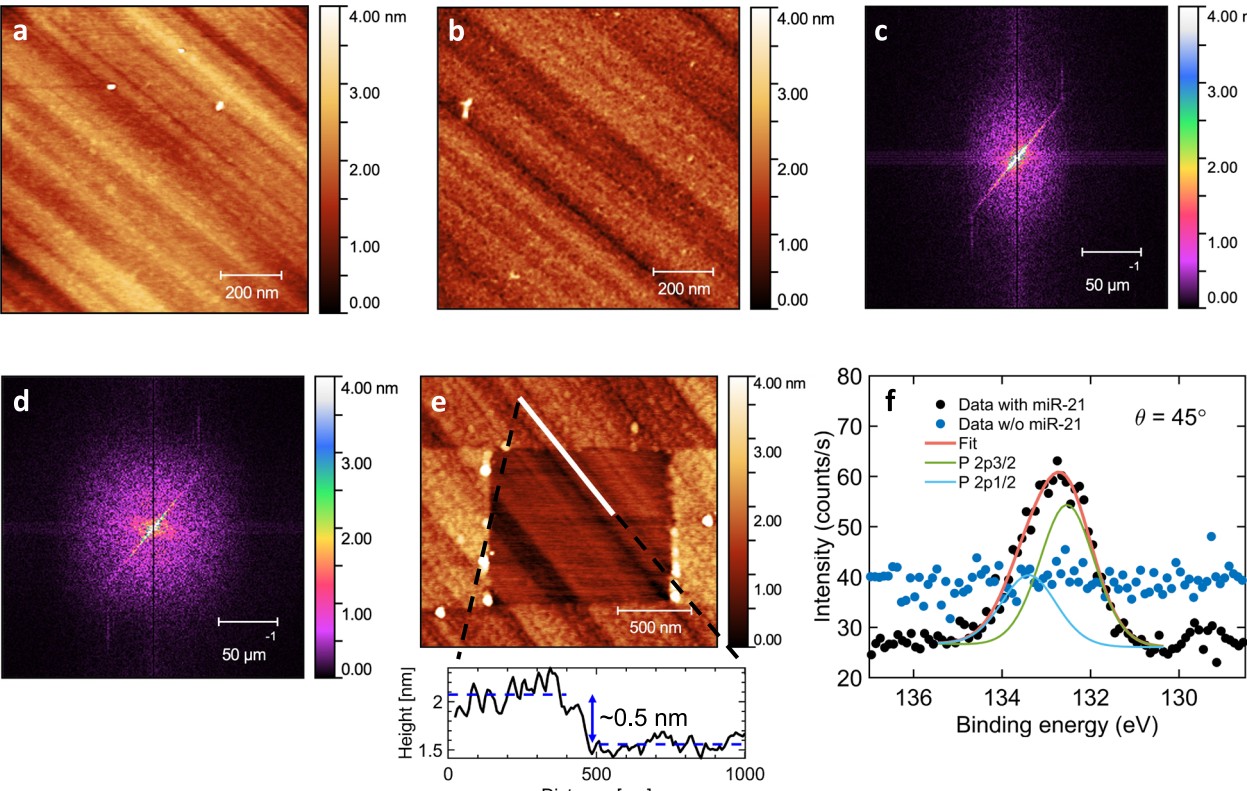

**Fig. 3 | Atomic force microscopy (AFM) and X-ray photoelectron spectroscopy (XPS) of miR-21 adsorbates. a** The AFM image of the diamond surface topography after treatment with Piranha solution. Diagonal stripes represent marks from the polishing process. **b** The AFM image of the diamond surface topography after exposing surface to miR-21. Polishing stripes are still visible but appear less sharp. **c** 2D Fourier transform (2DFFT) of the AFM image depicted in **a**. The bright diagonal line originates from the polishing marks. Less visible horizontal and vertical bands are due to the scanning induced artefacts. The diffuse halo around the central spot represents fine topographical features present on the surface. **d** 2DFFT of the

AFM image depicted in **b**. The central halo has increased in diameter, compared to **c**, corresponding to the presence of the added high frequency topographical features due to the adsorbed miR-21. **e** The AFM image of the diamond surface topography after scraping miR-21 with the AFM tip. The white line represents the data points used to estimate the height profile shown in the inset. The blue dashed lines represent the average heights for the scraped and unscraped sides. **f** High-resolution P 2p XPS spectra before and after the miR-21 treatment of the diamond surface. The P 2p peak originating from miR-21 phosphate groups is clearly visible for the treated surface and is absent in the nontreated sample.

the width of the single stranded microRNA indicating the presence of the single layer of miR-21 on the surface of diamond.

To unambiguously confirm adsorption of miR-21 we perform XPS analysis of the diamond surface exposed to miR-21. After measuring the Piranha treated diamond surface, the sample was treated with miR-21 in the same way as for the AFM measurements and remeasured with XPS. Figure 3f shows the high-resolution P 2p XPS spectra before and after the miR-21 treatment, confirming the adsorption of miR-21 on the diamond surface by the clearly visible P 2p peak originating from miR-21 phosphate groups. The presence of a Mn 2p peak in the survey spectrum (see Supplementary Fig. S5c) supports the proposed mechanism in which Mn ions mediate the adsorption of miR-21 onto the oxygenated diamond surface.

**Molecular dynamics simulations**

To better understand the experimental results, we perform all-atom molecular dynamics (MD) simulations of single miR-21 interacting with Mn ions and functionalized diamond surface. Figure 4a shows the simulation setup consisting of the capped box, two slabs of diamond along the $z$-axis and one miR-21. The volume between the diamond slabs is filled with aqueous solution having the same $Mn^{2+}$ concentration as in the experiments, and each diamond surface is terminated with hydroxyl/epoxy, carbonyl, ether, and carboxyl functional groups using estimates from the XPS measurements (see Supplementary Notes 1 and 2). Thanks to the two identical diamond slabs in the setup, the comparison of ion accumulation in the presence and in the absence of miR-21 adsorption is directly available within each simulation.

We simulate six systems differing in the composition (either hydroxyl or epoxy groups) and charge of the diamond surface (neutral or negatively charged, by deprotonation of the COOH). Such choice helps to separate and study the various effects of miR-21 and $Mn^{2+}$ adsorption and to account for the various possible surface compositions compatible with the XPS measurements. We start the simulations (48 replicas for 200 ns for each system) with the miR-21 placed in the middle between the two slabs. As the simulations progress, the miR-21 moves around the box and, in a subset of replicas, it eventually contacts the diamond surface. After an adsorption event takes place, we observe no detachment from the surface in the rest of the simulated time. Adsorption can occur on either the bottom or top surface. For subsequent analyses, we consider the $z$-coordinate as the distance from the diamond slab involved in the adsorption.

The percentage of replicas exhibiting miR-21 adsorption varies depending on the surface charge and composition (see Supplementary Table S3). Simulations show that for electrostatically neutral surfaces (either with epoxy or hydroxyl groups), the adsorption rate is approximately 80% and it is around 40% and 50% for the charged surface having either epoxy or hydroxyl groups, respectively. Furthermore, adsorption is equally stable for either of the charge states of the diamond surface. However, adsorption occurs faster for the neutral surface i.e., when there is no electrostatic repulsion between miR-21 and the diamond surface. The statistics of adsorption events is reported and detailed in Supplementary Note 3.

Figure 4b shows the density profiles of Mn ions along the $z$-coordinate, generated by averaging the data from the subset of MD simulations exhibiting miR-21 adsorption. Comparing the profile on the adsorption side

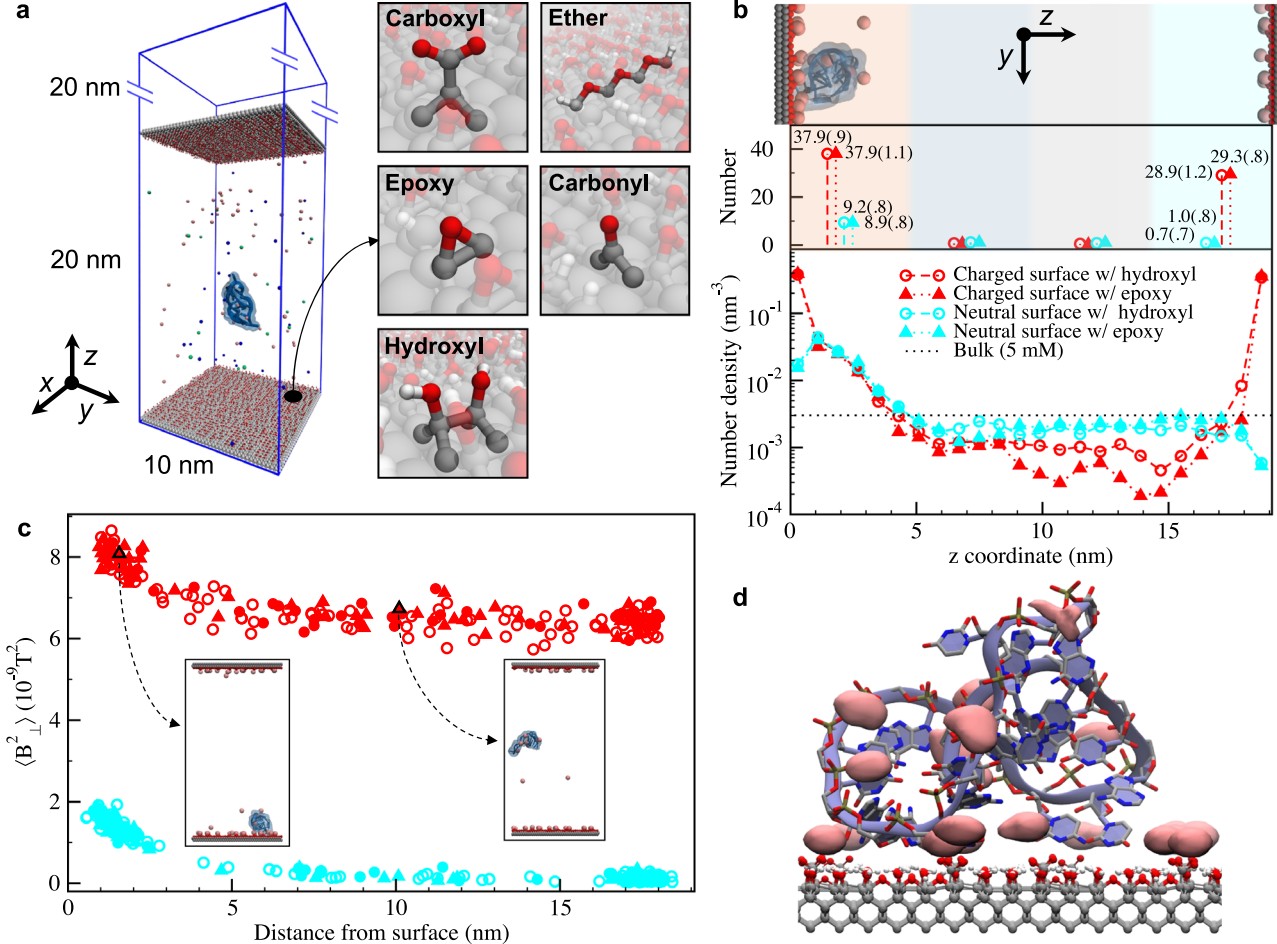

**Fig. 4 | Molecular dynamics (MD) simulations. a** Simulation setup consisting of capped box, two slabs of diamond along the $z$-axis and single miR-21. The volume between the diamond slabs is filled with aqueous solution and each diamond surface is terminated with hydroxyl/epoxy, carbonyl, ether, and carboxyl functional groups. **b** Number density profiles of Mn ions along the $z$-axis when miR-21 is adsorbed onto the negatively charged and neutral diamond surface. The top panel shows a representative snapshot to help visualize the arrangement in the simulations analyzed in the underlying graphs. The density profiles in the bottom panel are integrated in the middle panel for each of the four colored regions with the total ion numbers explicitly reported. Standard deviations are reported in parentheses. **c** The average $\langle B_\perp^2 \rangle$ experienced by the NV at a 7 nm depth as a function of the distance between the miR-21 center of mass and the diamond surface for negatively charged and neutral diamond surface states as in **b**. Each data point corresponds to one of the 48 replicas and shows the average value for the 1 ns simulation. **d** The isosurfaces (salmon pink) of $Mn^{2+}$ number density calculated for one representative MD trajectory. Each subregion corresponds to an ion interacting either with the diamond surface, with miR-21 or with both.

(0–4 nm) with that on the opposite side (15–19 nm), it is apparent that adsorption of miR-21 leads to accumulation of Mn ions. This effect is particularly pronounced for the neutral surface, which, on the side without miR-21 (right side in Fig. 4b) adsorption, is depleted of $Mn^{2+}$ with respect to the density in the bulk, due to the presence of a water adlayer at the interface[54]. In the case of charged surface, the accumulation is seen as a shoulder around 1-3 nm, whereas the peaks result from the interaction of $Mn^{2+}$ with the negatively charged carboxyl groups on the surface. We observe that one miR-21 adsorbed to the diamond surface accumulates on average 8-9 Mn ions independently of the surface charge and composition (Fig. 4b, middle panel).

Figure 4c shows the $\langle B_\perp^2 \rangle$ (computed as reported in the Methods section) experienced by a NV at a 7 nm depth, as a function of the distance between the miR-21 center of mass and the diamond surface. When the nucleic acid molecule approaches the surface, there is a clear increase of magnetic noise in the range $1.2\text{-}1.5 \times 10^{-9}\text{T}^2$. We can extrapolate this estimate to the situation where the surface is completely covered by miR-21 (see Supplementary Note 4) and obtain a value of the order of $10^{-8}\text{T}^2$ for the enhancement of magnetic noise intensity $\Delta\langle B_\perp^2 \rangle$. To connect this value to the variation of NV relaxation rate $\Gamma_1$ from Eq. 1 one needs an estimate of $\tau_c$, the autocorrelation decay time of the $Mn^{2+}$ electronic spins. Unfortunately,

no independent measurement of this quantity is available for Mn ions bound to RNA. Assuming $\tau_c$ is in the range between 10 ps and 1 ns (see Supplementary Fig. S1), then the reported estimate of the $\Delta\langle B_\perp^2 \rangle$ yields a $\Delta\Gamma_1$ (see Eq. 1) value between 2.3 and 23 kHz. Therefore, the theoretical estimate of $\Delta\Gamma_1$ is compatible with the measured value of $\Delta\Gamma_1 = 1.5$ kHz (see Supplementary Fig. S2).

## Discussion

All-atom MD simulations show that each adsorbed miR-21 can accumulate on average 8 Mn ions within ~4 nm from the diamond surface. Since it requires ~180 Mn adsorbates to generate a sizable signal per diffraction limited spot[30], the sensitivity of the presented quantum sensing modality approaches ~23 miR-21s. Furthermore, the method of detecting magnetic noise originating from counter ions could be applied to the microRNA microarray technology, which is based on nucleic acid hybridization between target molecules and their corresponding complementary probes[24].

The most common approach is to label target microRNAs with fluorescent dyes and detect light from the hybridized targets. Therefore, sensing paramagnetic ions countering biomolecular charges of target molecules would eliminate this key step in microRNA microarray experiment and even allow detection in non-transparent liquids. The sensitivity of

the method could be further improved by measuring NV spin relaxation time using pulse sequence with low-frequency noise cancellation[27] and engineering NVs closer to the surface.

We performed MD simulations to elucidate the molecular mechanisms underlying the observed NV relaxation changes. To this aim we simulated multiple systems with varying composition and charge of the diamond surface, guided by the XPS analysis. The computational modeling predicts that miR-21 interacts with the functionalized diamond surface, also in the presence of a negative surface charge, and that its adsorption leads to the recruitment of additional paramagnetic ions to the diamond surface, in a number which is rather robust with respect to variation of the surface model and miR-21 conformation. At this current "proof of principle" stage, we can only provide a crude estimate of the increase in NV relaxation rate, based on several assumptions and approximations. However, the predicted order of magnitude (kHz) of the change agrees with the experimental measurements.

In summary, we have demonstrated quantum sensing of miR-21 using NV centers in diamond by measuring magnetic noise of paramagnetic Mn ions counteracting negative charges of the nucleic acids. We have showed that such quantum sensing modality can be used to detect the presence of nucleotides close to the diamond surface avoiding limitations associated with the Debye screening. Differing from other methods, which require labels or labor-intensive amplification steps, sensing of magnetic noise can provide a required sensitivity for label-free detection of microRNAs for diagnostics of diseases such as cancer. While microRNAs are merely special cases of polyelectrolytes, we stress that our method should be easily extendable to sensing of other types of polyelectrolytes (i.e., polyanions interacting with paramagnetic cations), both natural and synthetic. We therefore expect this method to find use across different scientific fields utilizing polyelectrolytes, such as water treatment, filtering, enhanced oil recovery, food science, cosmetics, batteries and, as demonstrated in this study, biomedical applications.

## Methods
### Sample preparation and characterization
The electronic grade single crystal diamond plate $2 \times 2 \times 0.5$ mm in size was purchased from Element Six and laser-cut into two thinner plates. One of the plates, further referred to as the diamond, was polished down to 100 µm in thickness with one of the sides having roughness (Ra) Ra <1 nm and the other side Ra <2 nm, both corresponding to {100} faces. The diamond was then boiled for ~2 h in equal parts of sulfuric ($H_2SO_4$) and nitric ($HNO_3$) acid (diacid) and the {100} face with Ra <1 nm was homogeneously implanted with diatomic nitrogen at a fluence of $10^{13}$ cm$^{-2}$ and energy of 4 keV. After implantation, the diamond was boiled for ~2 h in diacid and annealed in vacuum for 4 h at 800 °C yielding a shallow layer of NV centers distributed $7 \pm 3$ nm below the surface as estimated using SRIM simulations[55].

Before each experiment the diamond was ultrasonicated in acetone for 20 min followed by surface oxygenation to form hydroxyl (C-OH), carbonyl (C = O), epoxy (C-O-C), ether (O-C-O) and carboxyl (COOH) functional groups. This was achieved by treating the diamond for 30 min. in Piranha solution prepared by mixing sulfuric acid with hydrogen peroxide (7:3 $H_2SO_4$(97%):$H_2O_2$(31%)).

The composition of functional groups on the surface of the diamond was determined using X-ray photoelectron spectroscopy (XPS) performed in a Kratos Axis Ultra DLD system by using a monochromatic Al Kα X-ray source (1486.3 eV) operating at 10 kV and 10 mA. The measurements were performed at a photoelectron detection angle of 45°. Survey and regional scans were acquired with pass energy of 160 and 20 eV, respectively. The step size was set to 1 eV for the survey and 0.1 eV for regional scans. The reported spectra were charge corrected with reference to the diamond C1s peak, commonly reported in the literature to be 284.5 eV. Acquired data were analyzed by using CasaXPS (Casa Software Ltd). The high-resolution spectrum was fitted to symmetric peaks by a Gaussian (70%)/Lorentzian (30%) product formula, using a Shirley background. For the C1s spectra, the peaks corresponding to C-O functionalities were constrained to the same

full-width-at-half-maximum. After treatment of the oxygenated diamond surface with miR-21, owing to the complexity of the different C-containing layers, 3 C 1 s peaks were added and auto-fitted to the spectrum to include the different C-O and C-N functionalities at binding energies higher than 285 eV, while peaks corresponding to $sp^3$ diamond, defects and carbides were constrained to the same energy shifts, full-width-at-half-maximum and area ratios as in the control sample prior to miR-21 treatment. This allowed for the determination of the C 1 s $sp^3$ diamond peak position (referenced to 284.5 eV), which was used in the charge corrections of the rest of the spectra after miR-21 treatment.

**Preparation of solutions.** Stock solution of 5 mM of $MnCl_2$ and 10 mM of NaCl was prepared by dissolving manganese (II) chloride tetrahydrate (CAS: 13446-34-9, Merck) and diluting 5 M NaCl buffer (AM9760G, Thermo Fisher Scientific) in ribonucleases free, DEPC-treated water (CAS: 7732-18-5, Thermo Fisher Scientific). This stock solution was then split into two parts. One part was used for control measurements and the other part was used to prepare microRNA solutions. Single-stranded miR-21 (sequence: 5′-UAGCUUAUCAGACUGAUGUUGA-3′) with no additional end or internal chemical modifications were purchased from Biomers.net GmbH in a dried state. MiR-21 was then dissolved in 5 mM $MnCl_2$ and 10 mM NaCl stock solution to obtain 1 µM and other concentration microRNA solutions. Tris-EDTA buffer solution (SKU: 93283-500 ML, Merck) was used to flush diamond surface prior to sensitivity measurements. To investigate dependence of the diamond surface charge state on the pH of the solution we split 5 mM $MnCl_2$ stock solution into four parts. The pH values for the three solutions were lowered down to 4, 3, and 2 using 1 M hydrochloric acid (HCl) solution while pH of the fourth solution was not adjusted.

**Experimental setup.** Interactions of miR-21 with the diamond surface were studied using an in-house wide-field microscope. Continuous wave 532 nm laser (Opus 532, Laser Quantum) was used to excite NV ensemble. The laser light was directed through the acousto-optic modulator (ISOMET, model M1133-aQ80L-1.5) and focused via dichroic mirror (DMLP550, Thorlabs) onto the back aperture of the 20×, 0.75 NA microscope objective (Nikon) allowing to probe ~2300 µm² area on the diamond surface. The excitation power density of 9 kW cm$^{-2}$ was kept the same for all the measurements. Fluorescence light from NV centers was filtered with a long-pass edge filter (LP02-633RU-25, Semrock) and projected onto a Si avalanche photodiode (APD410A, Thorlabs). Inspection of the diamond surface and focus adjustment were performed using digital CMOS camera (ORCA-Flash 4.0, Hamamatsu). The timing of light pulses was controlled with a 500 MHz PulseBlaster card (ESR-Pro-II, Spincore). Digitization and acquisition of analog data from the Si photodiode were performed with a multifunction input/output card (PCIe-6323, National Instruments) via a coaxial terminal block (BNC-2110, National Instruments). All experiments were run using a modified Python package described in ref. 43.

After surface functionalization the diamond was glued using an ultraviolet (UV) curing adhesive (Norland Blocking Adhesive 107, Norland Products) to a microscope cover glass (22 × 22 mm No. 1, Thermo Scientific) by UV light exposure for 5 min. in a curing chamber (ELC-500, Electro-Lite Corporation). The cover glass with the diamond was then mounted to a microfluidic device with a simple one-in-one-out port configuration (see Fig. 1a). The microfluidic channel that hosted the diamond had a volume of ~12 mm³ and was made in polydimethylsiloxane (PDMS) by casting onto a negative specially designed master mold. A microfluidic flow controller (OB1 MK3 + , Elveflow), a 10-channel distributor valve (Distributor MUX, Elveflow) and a flow sensor (FS4D, Elveflow) were used for sequential liquid injection into the microfluidic device and flow control.

**Spin relaxation measurements.** We performed all-optical longitudinal spin-lattice relaxation time $T_1$ measurements of NV centers with no

external magnetic field applied. Each data-point of $T_1$ curve is an average of 20,000 measurements performed using the pulse sequence depicted in Fig. 1b with the laser pulse duration set to 5 μs. The spin contrast was estimated from a ratio of two single point measurements at time $\tau_1 = 10$ μs and $\tau_2 = 400$ μs. Interactions of the diamond with miR-21 were probed at a constant flow rate of 50 μL min$^{-1}$ while flow rate for the other fluids were kept at 150 μL min$^{-1}$ during the measurements if not stated otherwise.

To determine the sensitivity of the presented method, the number of measurements was doubled to $N = 40,000$ for each interrogation time τ. Each data-point of the spin relaxation contrast ($C$) was temperature-corrected so all the corrected values ($C^*$) corresponded to the spin contrast as measured at the same temperature of 25 °C. The temperature-corrected $C^*$ is given by $C^* = C - (T - 25)\frac{\Delta C}{\Delta T}$, where $\frac{\Delta C}{\Delta T} = 0.0012$ °C$^{-1}$ (see Supplementary Fig. S3).

**Atomic force microscopy (AFM)**. The presence of miR-21 on the diamond surface was investigated with AFM (Asylum Research, Jupiter XR) using a wear resistant diamond coated tip (Adama AD-2,8-SS). Images for the untreated and exposed to miR-21 diamond sample were acquired in the tapping mode in the repulsive force regime. The AFM tip stayed mounted while imaging both surfaces and the scanning parameters remained unchanged. To cover the surface with miR-21, the diamond sample was treated in a solution containing 1 μM of miR-21 in 5 mM MnCl$_2$ and 10 mM NaCl for 5 min., rinsed with DEPC-treated water and dried using compressed CO$_2$. The diamond surface, covered with miR-21, was scraped using the same AFM tip in the contact mode using a force of 230 nN. Subsequently, the image mode was changed back to the tapping mode and a larger scan size was used to image the scraped and unscraped areas together. The images were analyzed using Gwyddion software.

**Simulation setup**. The simulated system (see Fig. 4a) was a $10 \times 10 \times 20$ nm$^3$ box containing the miR-21, water molecules and ions (Mn$^{2+}$, Na$^+$, Cl$^-$). The box was capped on both sides along the $z$-axis by two slabs of suitably functionalized diamond. Periodic boundary conditions were applied with a $z$-axis length of 40 nm for the elementary cell. This choice introduced an empty buffer between images along the $z$-axis, minimizing electrostatic effects caused by the accumulation of charges on one side of the slab. Our models for the diamond surface included hydroxyl/epoxy, carbonyl, ether, and carboxyl functional groups in numbers reproducing the estimates from the XPS measurements. The XPS measurements cannot differentiate between epoxy and hydroxyl groups, therefore, we used two sets of models: one containing only epoxy groups and the other containing only hydroxyl groups. The total surface charge was modulated by protonating half or all the carboxyl groups (see Supplementary Table S1). Each model featured identical top and bottom surfaces, enabling a direct comparison between the effects of miR-21 adsorption and the absence thereof.

The number of each ionic species in simulations (see Supplementary Table S1) was chosen to represent the bulk conditions in experiment ([MnCl$_2$] = 5 mM and [NaCl] = 10 mM). Additional cations were included to neutralize the negative charge of the miR-21 (-21$e$) and of the surfaces, when charged (either -65$e$ or -32$e$). Regarding the surfaces, we used the analytical solution to the Poisson-Boltzmann equation for a 1:1–2:1 electrolyte in the presence of a uniformly charged plane[56]. Regarding the ions around miR-21 we performed preliminary simulations of miR-21 in a box of water molecules and extracted the number of ions. The details of each model are reported in Supplementary Note 1.

**Molecular dynamics simulations**. The AMBER ff14SB-bsc1 was used for miR-21[57] together with the TIP3P water model[58], the Joung and Cheatham parameter set for monovalent ions[59] and the Li and Merz set for Mn$^{2+}$ [60]. The force field parameters for the diamond (see Supplementary Table S2) were derived based on density functional theory

(DFT) calculations as reported in the Supplementary Note 3. The MD simulations and data analysis were performed with GROMACS[61].

We performed preliminary replica-exchange molecular dynamics (REMD) simulations with temperatures from 298 K to 348.5 K of solvated miR-21 (see Supplementary Note 1 for further details) to obtain starting configurations for the subsequent simulations. Each of these miR-21 configurations was placed roughly in the middle of the simulation box of Fig. 4a, and for each system, we ran a REMD simulation with 48 replicas in the 298–337 K temperature range. This choice enabled a faster sampling of the configurational landscape of miR-21 and of the adsorption events.

A 4 fs timestep was used in the MD simulations, thanks to the employment of the hydrogen mass repartitioning scheme[62]. The volume was kept fixed while temperature was controlled with the v-rescale thermostat[63] with 0.2 ps coupling time. Particle Mesh Ewald was used to treat electrostatic interaction within periodic boundary conditions, with a cutoff of 1 nm for the Coulomb interaction. We used Ewald sum in three dimensions with a correction term (setting *ewald-geometry = 3dc* in Gromacs), which is appropriate when a vacuum buffer is present between the slab images[64]. A replica exchange was attempted every 200 steps. Each system was simulated for 200 ns after a 1 ns equilibration step with no exchange among the replicas. After the 200 ns runs, an additional 1 ns simulation was performed for each replica without exchange, to calculate density profiles and magnetic noise (see below and Supplementary Figs. S8 and S9).

**Magnetic noise estimation**. We estimated the magnetic noise by averaging the square of the magnetic dipole field of each Mn ion at the position of the NV center during the simulation and mediating over all possible spin states of the paramagnetic ions. The components of the magnetic field can be written as:

$$\langle B_k^2(\vec{\mathbf{r}})\rangle = C_S \left\langle \sum_{I \in \mathrm{Mn}} \frac{1 + 3\hat{k}_I^2}{r_I^6} \right\rangle,$$

$$C_S = \frac{S(S+1)}{3}\left(\frac{\mu_0}{4\pi}\gamma_e\hbar\right)^2 = 1.00574 \times 10^{-5}\ \mathrm{nm^6 T^2}$$

where $r_I$ is the distance between the $I$-th Mn$^{2+}$ and the NV center and $\hat{k}_I^2 = k_I^2/r_I^2$ with $k = x, y, z$ indicating the vector components (see Supplementary Note 4 for details). The $S = 5/2$ is the spin of Mn$^{2+}$, $\gamma_e$ is the electron gyromagnetic factor and $\mu_0$ is the vacuum permittivity. The average component orthogonal to the NV center orientation in the present case of a {100} diamond surface is given as:

$$\langle B_\perp^2 \rangle = \frac{2}{3}\left(\langle B_x^2 \rangle + \langle B_y^2 \rangle + \langle B_z^2 \rangle\right).$$

These calculations were performed using a purpose-made python script employing the MDAnalysis package[65]. The NV-center was placed at 7 nm below the surface. Due to the different nature of the various functional groups, the coordinate corresponding to 0 distance from the surface is better defined starting from the topmost layer of bulk carbon atoms, which are disposed in a virtually perfect plane. We thereby set the 0 of the surface 0.2 nm away from this layer, which corresponds to the average position of the epoxy oxygen atoms (see Supplementary Fig. S7). The $x$ and $y$ coordinates of the NV were aligned with the center of mass of miR-21, to reflect experimental conditions where the diamond surface is presumably entirely covered with nucleic acid molecules. During the 1-ns MD runs over which the average is taken, the miR-21 center of mass moved only negligibly. To increase the statistics, we considered NV centers in both the bottom and the top diamond slabs. We observe that other choices for the NV center placement would not qualitatively change our conclusions, and that more accurate predictions will need to consider other factors, such as the distribution in NVs implantation depth.

## Data availability

Data supporting the findings of the current study are available within the article and its Supplementary Information, and from the corresponding authors upon request.

## Code availability

The computer code used to generate results reported in the current study is available from the corresponding authors upon request.

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

## Acknowledgements

The Research Council of Norway is acknowledged for the support of the Norwegian Micro- and Nano-Fabrication Facility, NorFab, project number 295864. The authors acknowledge the use of the LUMI HPC facility within project n. 465000458 and of the ISCRA-CINECA facility (Bologna) within the ISCRA-C 2022 project HP10CQEFNU. This research was supported by Next Generation-EU through the project Tuscany Health Ecosystem (THE-Spoke 1, grant ECS_00000017) and by the European Research Council under the EU-H2020 FETPROACT Programme (LESGO project, Agreement No. 952068).

## Author contributions

J.Z. conceived the idea, designed the experiment, performed spin relaxation measurements, and took part in the writing of the manuscript. M.M.G. performed AFM measurements, analyzed the data, and took part in the writing of the manuscript. V.H. performed XPS measurements and analyzed the data. V.T. and G.S. participated in the design and interpretation of simulations and the manuscript check. L.B. performed the DFT calculations and participated in the design and interpretation of the simulations R.N. participated in the design and interpretation of the simulations, performed simulations, analyzed results, and took part in the writing of the manuscript.

## Funding

## Competing interests

The authors declare no competing interests.
