## [Peer Review File · Communications Chemistry]

Reviewers' comments:

Reviewer #1 (Remarks to the Author):

The manuscript authored by Zalieckas et al. treats a novel and very interesting topic – the utilization of NV-centers in diamond for the detection of microRNAs. Instead of focusing on detecting the electric field of the charged backbone, they propose a pioneering approach where paramagnetic ions interacting with the microRNA are detected using NV-relaxometry. While I find the concept of the work appealing, I have several concerns regarding the data and the overall message. Addressing these concerns would be important for publication in CommsChem. Alternatively, considering a more specialized journal (e.g., J. Phys Chem series) might also be appropriate.

Major Comments:

- The authors employ a microwave-free approach to measure T1 relaxation. Recent studies have indicated that these experiments may be susceptible to various influences such as NV charge state (ref: <https://arxiv.org/abs/2301.01063>), temperature (ref: <https://arxiv.org/abs/2309.12131>), diamagnetic ions (ref: <https://pubs.acs.org/doi/abs/10.1021/acsnano.3c01298>), or electric field noise (ref: <https://doi.org/10.1103/PhysRevLett.118.197201>). For instance, the adsorbed RNA might induce electrical noise or alter the local NV charge state, potentially leading to similar experimental outcomes. This could be investigated by repeating the experiments using RNA without Mn²⁺. While acknowledging that the authors conduct relative measurements and observe distinct effects, it is necessary to discuss potential sources of their observed signal.
- The definition of contrast lacks clarity. It is unclear why they chose $\tau_1 = 10$ microseconds and why not $\tau_1 = 0$ microseconds. Please clarify.
- Figure 2b, pH Dependence: Considering the known influence of pH on T1 time (ref: <https://pubs.acs.org/doi/abs/10.1021/acsnano.9b05342>) where lowering the pH typically increases T1. For that reason, it is expected that the authors observe two conflicting mechanisms. Moreover, the T1 contrast at pH2 should be lower compared to pure water (H₂O). Further clarification is necessary.
- The demonstration of an effective RNA sensor seems insufficient in the presented work. The change in contrast as a function of concentration in Figure 1D appears minimal. It would be helpful to visualize the signal response for concentrations like 1 fM or 1 mM to establish a calibration curve if the focus of the work is on sensing applications. Additionally, a discussion of practical applications remains unaddressed (e.g., how distinguish different sequences?). The more intriguing aspect appears to be the adsorption of molecules on the diamond surface, a fact that has not been extensively explored prior to this work.
- Figure 4b: It is not fully clear to me why the charged surface has more ions at the surface compared to the neutral surface (blue vs red). Clarification is needed regarding why the blue curve diminishes at the interface.

Minor Comments:

- The figures should follow the sequence in which they are discussed in the manuscript.
- Additional experimental conditions within the T1 figures would enhance clarity and comprehension. For instance, including “Mn²⁺” alongside pH values or miR-21 in Figure 2b/1d.

Reviewer #2 (Remarks to the Author):

The manuscript by Justas Zalieckas and co-workers represents a pioneering exploration into the utilization of nitrogen-vacancy (NV) centers in diamond for the detection of nucleic acids in solution, with a particular focus on microRNA-21 (miR-21). The ability to monitor and quantify the presence of molecules such as miR-21 is of fundamental importance finding potential application in diverse scientific fields, such as nanotechnology and biomedicine. Therefore the subject of the manuscript is of great relevance for the readership of Communications Chemistry.

The authors propose to detect the presence of miR-21 by monitoring the change in the longitudinal spin-lattice relaxation time of the NV center due to magnetic noise induced by the ionic "cloud" of manganese atoms that surround the miR-21. Experimental measurements show a clear change in the fluorescence contrast in the presence of miR-21. Furthermore, to corroborate that the observed change originates from the presence of miR-21 on the diamond surface, atomic force microscopy (AFM) measurements were conducted. Although AFM is unable to directly detect the presence of miR-21, the authors' analysis of the variation in the surface profile before and after adsorption is compelling. Additionally, molecular dynamics (MD) simulations have been performed to investigate the adsorption mechanism and the consequent distribution of Mn ions. From these data, an estimate of the magnetic noise on the measured longitudinal relaxation-time has been extracted. The authors report a variation in the computed longitudinal relaxation time of about 5 kHz, and they compare this value with the measured one. Since the observed relaxation times are presented as contrasts in the experimental section, I believe that extracting a value directly comparable with the theoretical estimation would provide a much clearer indication of the accuracy of the simulations. Additionally, it would facilitate future efforts to reproduce such values.

The authors support their conclusions with enough data, which are clearly arranged and explained. I support the publication of the manuscript in Communications Chemistry.

A. Lorenzo Mariano (Trinity College Dublin)

Reviewer #3 (Remarks to the Author):

The manuscript by Zalieckas describes the use of NV ensembles to detect the presence of mRNA close to diamond surfaces via their association with paramagnetic Mn(2+) ions and resultant drop in T1 relaxation time of the shallow NV defects. This work adds to the growing number of reports in the exciting field of using NV quantum sensors for chemical sensing applications, and addresses a particular advantage the NV systems could offer, namely overcoming Debye length limitations for sensing charged

biomolecules. While detection of Mn(2+) ions by NV centers has been reported previously (refs 30,31 main text), this manuscript demonstrates nicely the dynamic sensing capabilities in a flow-cell design to probe oligonucleotides in a solution. The manuscript is well-written and the results are convincing and are supported by proper control experiments. Molecular dynamics simulations are used to complement the experimental findings by modeling binding interactions between mRNA molecules and the diamond surface with a variety of plausible surface terminations. I recommend for publication following addressing of the following points:

1. The XPS survey scan in the supporting information shows significant contamination by what I assume is silicon on the diamond surface. I believe the authors should address this point and comment on how that may influence Mn association with the surface and/or mRNA binding.
2. XPS analysis of the surface after binding of the mRNA should be done. Observation of phosphorous 2p signals would give unambiguous chemical identification of mRNA adsorption to support the conclusions drawn from the AFM experiments. Such an XPS analysis could also compare conditions of mRNA adsorption in the presence or absence of Mn ions.
3. Figure 1d shows the results of 1 micromolar versus 1 nanomolar additions of the mRNA solution in the presence of Mn(2+) ions. These results should be analyzed in greater detail to evaluate more clearly the sensing ability of the NV platform to distinguish different analyte concentrations. Is there a statistically significant difference between the measured contrast values for these two mRNA concentrations shown? Ideally, a dose-response curve should be created with multiple concentrations to determine the most sensitive detection range.
4. Oligonucleotide concentrations are used that are orders of magnitude smaller than the concentration of Mn(2+) ions. At this high ratio of Mn to phosphate groups in the mRNA, aggregation of the oligonucleotides can occur (DOI: 10.1093/nar/gkh242) which would have a significant impact on how the molecules are interacting with the diamond surface. Can the authors address this? Could this be why there is not significant difference in contrast measured between 1 micromolar and 1 nanomolar concentrations of mRNA? Perhaps CD or UV-Vis measurements of the oligos with different concentrations of Mn.
5. How is the measured T1 of the NV ensembles influenced by exposure to the mRNA without the Mn?

Dear Reviewers,

We would like to thank you for your comments and suggestions. We believe that we addressed all of them and, as a result, it made the manuscript much better. Below we provide answers to your questions.

Reviewer #1 (Remarks to the Author):

The manuscript authored by Zalieckas et al. treats a novel and very interesting topic – the utilization of NV-centers in diamond for the detection of microRNAs. Instead of focusing on detecting the electric field of the charged backbone, they propose a pioneering approach where paramagnetic ions interacting with the microRNA are detected using NV-relaxometry. While I find the concept of the work appealing, I have several concerns regarding the data and the overall message. Addressing these concerns would be important for publication in CommsChem. Alternatively, considering a more specialized journal (e.g., J. Phys Chem series) might also be appropriate.

Major Comments:

• *The authors employ a microwave-free approach to measure T1 relaxation. Recent studies have indicated that these experiments may be susceptible to various influences such as NV charge state (ref: <https://arxiv.org/abs/2301.01063>), temperature (ref: <https://arxiv.org/abs/2309.12131>), diamagnetic ions (ref: <https://pubs.acs.org/doi/abs/10.1021/acsnano.3c01298>), or electric field noise (ref: <https://doi.org/10.1103/PhysRevLett.118.197201>). For instance, the adsorbed RNA might induce electrical noise or alter the local NV charge state, potentially leading to similar experimental outcomes. This could be investigated by repeating the experiments using RNA without Mn²⁺. While acknowledging that the authors conduct relative measurements and observe distinct effects, it is necessary to discuss potential sources of their observed signal.*

We thank the reviewer for their positive appraisal of the concept behind our work. As suggested, we included a discussion on potential sources (NV center charge conversion, diamagnetic electrolyte solutions, electric noise, and temperature) to the signal in the “Results” section. Since the excitation power intensity (9 kW/cm²) used in this work is below the saturation intensity (~100 kW/cm²) and surface oxygenation stabilizes shallow negatively charged NVs (<https://doi.org/10.1021/acs.jpcc.8b11274>), effects attributed to NVs charge conversion are neglected in this work. We also do not expect to see any effect on T1 due to diamagnetic ions since we use 10 mM NaCl concentration, which was shown not to affect T1 (<https://pubs.acs.org/doi/abs/10.1021/acsnano.3c01298>). Since T1 is susceptible to temperature (see Supplementary Fig. S3), we temperature-corrected the data-points for new sensitivity measurements included in the revised manuscript (Fig. 1d) to account for temperature fluctuations in the room during the measurements. All above-mentioned influences have negligible effect on the main result presented in Fig. 1c. We also tested if charged miR-21 adsorbates induce observable electric noise by measuring the change in spin relaxation contrast with no Mn ions present (see Supplementary Fig. S4). We observed no

measurable change in the spin contrast demonstrating that electric noise has negligible effects on the presented measurements.

- *The definition of contrast lacks clarity. It is unclear why they chose $\tau_1 = 10$ microseconds and why not $\tau_1 = 0$ microseconds. Please clarify.*

In the experimental setup, electronic and other components such as acousto-optic modulator have finite response times (rise/fall times). Therefore, it is technically difficult to set τ_1 strictly to 0 μs and avoid potential overlaps. Instead, it is set to 10 μs as was done in previous works (10.1038/srep22797 (2016)). We clarified this in the “Results” section.

- *Figure 2b, pH Dependence: Considering the known influence of pH on T1 time (ref: <https://pubs.acs.org/doi/abs/10.1021/acsnano.9b05342>) where lowering the pH typically increases T1. For that reason, it is expected that the authors observe two conflicting mechanisms. Moreover, the T1 contrast at pH2 should be lower compared to pure water (H2O). Further clarification is necessary.*

In ref. <https://pubs.acs.org/doi/abs/10.1021/acsnano.9b05342> authors use nanodiamonds with uniform surface carboxylation to probe T1 dependence on pH. In our work, we measured that only 1.6% of the diamond surface is covered with COOH groups. Therefore, in ref. <https://pubs.acs.org/doi/abs/10.1021/acsnano.9b05342> the dominant contribution to T1 comes from the electric field generated by the negative charge while in our work, due to low COOH content, this contribution is expected to be very small. The main contribution to T1 in our work originates from the magnetic noise induced by Mn ion adsorbates. Hence, lowering pH protonates COOH groups, which attract less Mn ions thus lowering the spin contrast. Since the spin contrast is mostly affected by the magnetic noise from Mn ions, we expect that at pH 2 the spin contrast, due to absence of Mn ions, is at similar level as for pure water. Moreover, in Supplementary Fig. S4, where we tested if charged miR-21 adsorbates induce observable electric noise, we see no observable change in the spin contrast between solution I (pH 8.0) and solution III (pH 5.3). This confirms that the dominant effect to the T1 contrast originates from Mn ions induced magnetic noise. We added additional clarification in the “Results” section.

- *The demonstration of an effective RNA sensor seems insufficient in the presented work. The change in contrast as a function of concentration in Figure 1D appears minimal. It would be helpful to visualize the signal response for concentrations like 1 fM or 1 mM to establish a calibration curve if the focus the work is on sensing applications. Additionally, a discussion of practical applications remains unaddressed (e.g., how distinguish different sequences?). The more intriguing aspect appears to be the adsorption of molecules on the diamond surface, a fact that has not been extensively explored prior to this work.*

We performed additional measurements to estimate the sensitivity of the presented method (see Fig. 1d). We tested various miR-21 concentrations and determined the limit of detection to be 10 pM for given experimental conditions. A calibration curve showing the change in spin relaxation contrast as a function of miR-21 concentration is presented in Supplementary Fig. S4.

The aim of this work is to demonstrate the feasibility of the new approach and, therefore, study of distinguishing different sequences is left for the future work. Moreover, we present theoretical study of miR-21 nucleotides interactions with diamond surface groups in Supplementary Note 3.

- *Figure 4b: It is not fully clear to me why the charged surface has more ions at the surface compared to the neutral surface (blue vs red). Clarification is needed regarding why the blue curve diminishes at the interface.*

Fig. 4b shows the number density profiles of Mn ions along the z-axis when miR-21 is adsorbed onto the negatively charged and neutral diamond surface. It is expected, and simulations show, that Mn ions are adsorbed onto the negatively charged surface. When the surface is neutral Mn^{2+} does not experience electrostatic attraction, whereas the layer of high-density water (water adlayer) at the interface still remains, preventing direct interaction and thus adsorption. The drop in Mn concentration at the interface with the neutral surface is thus due to the presence of the water adlayer combined with the absence of electrostatic attraction. We added a sentence in the manuscript to point this out.

Minor Comments:

- *The figures should follow the sequence in which they are discussed in the manuscript.*

We fixed this.

- *Additional experimental conditions within the T1 figures would enhance clarity and comprehension. For instance, including "Mn²⁺" alongside pH values or miR-21 in Figure 2b/1d.* We changed the representation of the experimental conditions within the figures as suggested.

Reviewer #2 (Remarks to the Author):

The manuscript by Justas Zalieckas and co-workers represents a pioneering exploration into the utilization of nitrogen-vacancy (NV) centers in diamond for the detection of nucleic acids in solution, with a particular focus on microRNA-21 (miR-21). The ability to monitor and quantify the presence of molecules such as miR-21 is of fundamental importance finding potential application in diverse scientific fields, such as nanotechnology and biomedicine. Therefore the subject of the manuscript is of great relevance for the readership of Communications Chemistry.

The authors propose to detect the presence of miR-21 by monitoring the change in the longitudinal spin-lattice relaxation time of the NV center due to magnetic noise induced by the ionic "cloud" of manganese atoms that surround the miR-21. Experimental measurements show a clear change in the fluorescence contrast in the presence of miR-21. Furthermore, to corroborate that the observed change originates from the presence of miR-21 on the diamond surface, atomic force microscopy (AFM) measurements were conducted. Although AFM is unable to directly detect the presence of miR-21, the authors' analysis of the variation in the surface profile before and after adsorption is compelling. Additionally, molecular dynamics (MD) simulations have been performed to investigate the adsorption mechanism and the consequent

distribution of Mn ions. From these data, an estimate of the magnetic noise on the measured longitudinal relaxation-time has been extracted. The authors report a variation in the computed longitudinal relaxation time of about 5 kHz, and they compare this value with the measured one. Since the observed relaxation times are presented as contrasts in the experimental section, I believe that extracting a value directly comparable with the theoretical estimation would provide a much clearer indication of the accuracy of the simulations. Additionally, it would facilitate future efforts to reproduce such values.

We thank the reviewer for raising this issue and agree that a direct comparison of the relaxation times would be helpful to better connect the simulations and the measurements. This comment also gave us the opportunity to evaluate more critically the question of the paramagnetic spin decay time (τ_c).

The NV longitudinal relaxations times cannot be directly obtained from the available contrast data. We therefore performed new measurements of the full relaxation curves in three conditions (see Supplementary Fig. S2):

- a) absence of Mn
- b) presence of Mn only ([Mn]=5mM)
- c) presence of Mn ([Mn]=5mM) and miR-21 (1 μ M).

We used a bi-exponential fit to extract the values of T_1 . The bi-exponential law is the one commonly employed and is justified by the presence of a fast-decaying population of NV close to the surface giving rise to the shorter relaxation time $T_{1,short}$, see for example ref. [30] in the main text. We consistently find $T_{1,short} \ll T_{1,long}$ and around few tens of us.

The resulting T_1 (i.e. $T_{1,long}$) are a) 634 μ s, b) 449 μ s, c) 266 μ s. From these values, the increase of the relaxation rate Γ_1 upon addition of microRNA (from b to c) is 1.5 kHz. Given the uncertainties detailed below, we believe that this value is compatible with our previous estimate of $\Delta\Gamma_1 \simeq 5$ kHz. The various assumptions in extracting this estimate were:

- 1) the value of the electric spin relaxation τ_c : this is a crucial parameter that may strongly depend on the interactions between the paramagnetic ion and the surrounding molecules. Unfortunately, no independent measurement of τ_c is available when Mn^{2+} interacts with nucleic acids and/or the charged diamond surface. In the submitted version we assumed a value of .5 ns as an upper limit, based on the dipolar relaxivity estimated by Ziem *et al* (ref [30] in the manuscript) for solvated Mn^{2+} , which is however still derived from NV relaxometry measurements. From a survey of the literature, we found values of electric spin relaxation in the 10ps \div 1ns range (see for example Esteban-Gómez *et al* RSC Advances 2013, doi: 10.1039/C3RA45721D). We recall that

$$\Delta\Gamma_1 = \frac{3\gamma_e^2}{\omega_0} \frac{\omega_0\tau_c}{1 + \omega_0^2\tau_c^2} \Delta\langle B_{\perp}^2 \rangle \cong 5.16 \times 10^3 \text{ GHz } T^{-2} g(\tau_c) \Delta\langle B_{\perp}^2 \rangle$$

where $g(\tau_c)$ is plotted in Fig. S1. A 10ps \div 1ns range for τ_c corresponds to $g(\tau_c)$ of .05 \div .50 (1/2 being the maximum possible value of $g(\tau_c)$), resulting in

$$\Delta\Gamma_1 = (2.6 \div .26) \times 10^9 \Delta\langle B_{\perp}^2 \rangle T^{-2} \text{ kHz}$$

- 2) we assumed complete coverage of the diamond surface by the microRNA. A less dense coverage would result in lower values of $\Delta\langle B_{\perp}^2 \rangle$. This issue can be investigated by future molecular modelling studies.
- 3) the NV implantation depth: we assumed a value of 7nm, but actually the NVs are distributed at various distances from the surface. Higher values would result in smaller values of $\langle B_{\perp}^2 \rangle$ and, thus, smaller $\Delta\Gamma_1$. We decided to keep the 7nm value and will consider the impact of a depth distribution in future investigations.

In addition, other decay laws could be considered, including a stretched exponential, resulting in different values for the T_1 (see for example Vedelaar et al arXiv preprint arXiv:2211.07269, 2022). Also, this point will be dealt with in future investigation.

Given all these issues, we preferred to modify the part regarding the estimate of $\Delta\Gamma_1$, and decided to give a plausible range rather than a single value: $\Delta\Gamma_1 = (2.3 \div 23)kHz$.

The authors support their conclusions with enough data, which are clearly arranged and explained. I support the publication of the manuscript in Communications Chemistry.

Reviewer #3 (Remarks to the Author):

The manuscript by Zalieckas describes the use of NV ensembles to detect the presence of mRNA close to diamond surfaces via their association with paramagnetic Mn(2+) ions and resultant drop in T1 relaxation time of the shallow NV defects. This work adds to the growing number of reports in the exciting field of using NV quantum sensors for chemical sensing applications, and addresses a particular advantage the NV systems could offer, namely overcoming Debye length limitations for sensing charged biomolecules. While detection of Mn(2+) ions by NV centers has been reported previously (refs 30,31 main text), this manuscript demonstrates nicely the dynamic sensing capabilities in a flow-cell design to probe oligonucleotides in a solution. The manuscript is well-written and the results are convincing and are supported by proper control experiments. Molecular dynamics simulations are used to complement the experimental findings by modeling binding interactions between mRNA molecules and the diamond surface with a variety of plausible surface terminations.

We thank the reviewer for the positive evaluation of our work and careful reading of our manuscript.

I recommend for publication following addressing of the following points:

1. *The XPS survey scan in the supporting information shows significant contamination by what I assume is silicon on the diamond surface. I believe the authors should address this point and comment on how that may influence Mn association with the surface and/or mRNA binding.*

Indeed, we observe peaks Si 2s and Si 2p in the survey spectrum that indicate sample contamination with silicon either in the bulk or on the surface. We updated the corresponding figure in the Supplementary Information. Efforts to improve cleaning and handling of the diamond sample did not suppress the observed peaks. Therefore, we assume that silicon contamination is mainly in the bulk and we expect that it does not significantly influence interaction of Mn ions and miR-21 with the diamond surface. We address this point in the “Results” section.

2. XPS analysis of the surface after binding of the mRNA should be done. Observation of phosphorous 2p signals would give unambiguous chemical identification of mRNA adsorption to support the conclusions drawn from the AFM experiments. Such an XPS analysis could also compare conditions of mRNA adsorption in the presence or absence of Mn ions.

As suggested, we performed XPS analysis before and after miR-21 adsorption and observed phosphorous 2p signal. We updated Fig. 3 and included discussion in the “Results” section.

3. Figure 1d shows the results of 1 micromolar versus 1 nanomolar additions of the mRNA solution in the presence of Mn(2+) ions. These results should be analyzed in greater detail to evaluate more clearly the sensing ability of the NV platform to distinguish different analyte concentrations. Is there is a statistically significant difference between the measured contrast values for these two mRNA concentrations shown? Ideally, a dose-response curve should be created with multiple concentrations to determine the most sensitive detection range.

We addressed this by measuring the response of the experimental setup to the injection of miR-21 concentrations ranging from 10 pM to 10 nM (see Fig. 1d) and discussed the results in the “Results” section. From the measurements we estimated the limit of detection to be 10 pM for given experimental conditions. Given the microfluidic channel volume of 12 mm³, the limit of detection of 10 pM translates to 120 attomoles. A dose-response curve showing the change in spin relaxation contrast as a function of miR-21 concentration is presented in Supplementary Fig. S4.

4. Oligonucleotide concentrations are used that are orders of magnitude smaller than the concentration of Mn(2+) ions. At this high ratio of Mn to phosphate groups in the mRNA, aggregation of the oligonucleotides can occur (DOI: 10.1093/nar/gkh242) which would have a significant impact on how the molecules are interacting with the diamond surface. Can the authors address this? Could this be why there is not significant difference in contrast measured between 1 micromolar and 1 nanomolar concentrations of mRNA? Perhaps CD or UV-Vis measurements of the oligos with different concentrations of Mn.

As mentioned above, we additionally measured spin contrast for miR-21 concentrations down to 10 pM (see Fig. 1d). We observed gradual and linear increase in spin relaxation for 10 pM, 100 pM and 1 nM concentrations. This indicates, at least for these concentrations, that there is no significant aggregation of the oligonucleotides. However, for 10 nM we observed non-linear

response. This might be attributed to aggregation or to the changes of the local environment such as pH and charge state of the surface. We included this discuss in the “Results” section.

5. How is the measured T_1 of the NV ensembles influenced by exposure to the mRNA without the Mn?

We tested if charged miR-21 adsorbates induce observable electric noise by measuring the change in spin relaxation contrast with no Mn ions present (see Supplementary Fig. S4). We observed no measurable change in the spin contrast demonstrating that electric noise has negligible effect on the presented measurements.

REVIEWERS' COMMENTS:

Reviewer #1 (Remarks to the Author):

I thank the authors for their additional work. I recommend the publication as all my questions have been thoroughly answered.

Reviewer #2 (Remarks to the Author):

I believe that the points raised in the previous round of review have been satisfactorily addressed. Several new measurements and corrections to the original manuscript have been made, increasing the scientific value of the work.

Reviewer #3 (Remarks to the Author):

The authors thoroughly addressed the comments and suggestions by me and the other reviewers. With the additional experiments, characterizations, and text edits added to the manuscript, I can recommend publication in Communications Chemistry.